# Motivating Physical Activity via Competitive Human-Robot Interaction

**Boling Yang[1], Golnaz Habibi[2], Patrick E. Lancaster[1], Byron Boots[1], Joshua R. Smith[1]**
[1] University of Washington, [2] Massachusetts Institute of Technology

**Abstract:** This project aims to motivate research in competitive human-robot interaction by creating a robot competitor that can challenge human users in certain scenarios such as physical exercise and games. With this goal in mind, we introduce the Fencing Game, a human-robot competition used to evaluate both the capabilities of the robot competitor and user experience. We develop the robot competitor through iterative multi-agent reinforcement learning and show that it can perform well against human competitors. Our user study additionally found that our system was able to continuously create challenging and enjoyable interactions that significantly increased human subjects' heart rates. The majority of human subjects considered the system to be entertaining and desirable for improving the quality of their exercise.

**Keywords:** Competitive-HRI, Reinforcement Learning, Multi-agent System

## 1 Introduction

Competition is ubiquitous in the natural world [1, 2] and in human society [3, 4, 5]. Despite its universality, *competitive interaction* has rarely been investigated in the field of Human Robot Interaction, which has mainly focused on *cooperative interactions* such as collaborative manipulation, mobility assistance, feeding, and so on [6, 7, 8, 9, 10]. In some ways it is not surprising that competitive interaction has been overlooked: of course everyone wants a robot that can assist them; who would want a robot that thwarts their intentions? Yet, we also accept that human-human competition can be healthy and productive, for example in structured contexts such as sports. In this paper we explore the idea that human-robot competition can provide similar benefits.

We believe that physical exercise settings such as athletic practice, fitness training, and physical therapy are scenarios in which competitive HRI can benefit users. Physical exercise is essential to our physical and mental health. In addition, the quality of some physical rehabilitation is closely related to patients' commitment to exercising regularly. However, the major cause of poor adherence to physical exercise is due to a lack of motivation [11, 12, 13]. It has been shown that enjoyment, competition, and challenges are important motivational factors for the practice of physical exercise [14, 12]. In this project, our goal is to create a robot that can provide these motivations to human users via adversarial behaviors. The main contributions of this paper are listed below:

**Motivating Competitive-HRI Research.** We discuss how competitive interaction can influence people in a positive manner and the technical difficulties competitive-HRI tasks represent. This discussion motivates our proposed Fencing Game, a competitive physically interactive game, as an evaluation environment for competitive-HRI algorithms and user study.

**System Design and Implementation.** A highly generalizable robotic system is created to support our competitive-HRI research. A multi-agent reinforcement learning(RL) method is used to train a robot to play competitive games with multiple gameplay styles.

**Two User Studies.** Our first user study found that more than 80% of subjects found competitive interaction with our robot to be entertaining and desirable. Our system was able to provide challenging gameplay experiences that significantly increase human players' heart rates. Furthermore, we observed that human subjects who constantly explore different strategies tend to achieve higher

5th Conference on Robot Learning (CoRL 2021), London, UK.

rewards in the long run. A second user study demonstrated that an RL-trained policy made it significantly more challenging for subjects to make quantitative improvement in a long sequence of games compared to a carefully designed heuristic baseline policy. The RL-trained agent also appeared to be more intelligent to the human subjects.

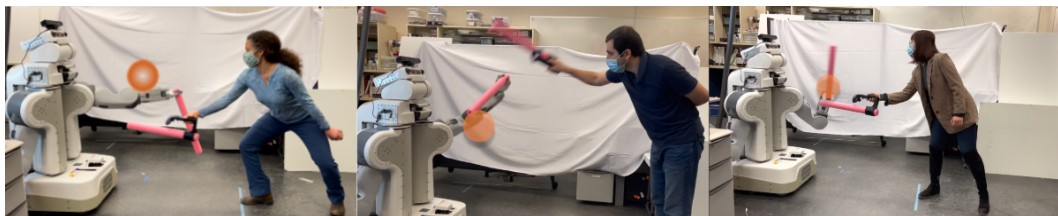

Figure 1: Competitive fencing games between a PR2 robot and human subjects. The detailed game rules are described in Sec. 2. Please refer to this link for example gameplay videos.

## 2 Competitive Interaction

In this section, we discuss the significance of competitive interaction and justify why we believe that it deserves increased attention in the robotics community. We will first discuss how competition can influence people in a positive manner from a psychological perspective. Afterward, we will discuss the technical challenges in competitive-HRI tasks and propose the use of the Fencing Game as the main task that this project will focus on.

**Positive Influences of Competitive Interaction.** Competition between players provides motivation and can foster improvement in performance at a given task. Plass et al. [15] compared competitive versus cooperative interactions in an educational mathematics video game. The results revealed that, compared to working individually, the subjects performed significantly better when working competitively. In particular, competitive players demonstrated higher effectiveness in problem solving compared to non-competitive players. This study also observed that subjects experienced a higher level of enjoyment via an increased tendency to engage in the game [16]. Furthermore, they also displayed higher situational interest, *i.e.* the subject paid more attention and had more interaction during the game [17]. Viru et al. [18] showed that competitive exercises can improve athletic performance in a treadmill running test. Subjects' average running duration was increased by 4.2%. During cycling and planking exercise, Feltz et al. [4] were also able to inspire higher performance from subjects by placing them in competition with a manipulated virtual partner.

Inspired by these studies, we envision that a personal robot can become a competitive partner that provides enjoyment, increases motivation, and motivates improvement in activities such as physical exercise. For this reason, we initiate our competitive-HRI research by focusing on creating a physical exercise companion.

**Technical Challenges.** Creating an actual robot that can compete with a human physically is challenging. The robot needs to constantly reason about the human's intent via their actions, and strategically control its high degree-of-freedom body to counteract the opponent's adversarial behavior and maximize its own return. Therefore, a big part of our competitive-HRI research focuses on solving these technical challenges to create a robotic system with real-time decision making capability and body agility that is comparable to that of humans.

**The Fencing Game.** Based on the expected technical challenges, we designed a two-player zero-sum physically interactive game. The Fencing Game is an attack and defense game where the human player is the antagonist with the goal of maximizing their game score. The robot is the protagonist who aims to minimize the antagonist's score. Fig. 1 shows three images of human subjects playing the game, and Algo. 2 in Appendix A.1 summarizes the scoring mechanism for this game. The orange spherical area located between the two players denotes the target area of the game. The antagonist on the right earns 1 point for every 0.01 seconds that their bat is placed within the target area without contacting the opponent's bat. The antagonist will lose 10 points if the antagonist's bat is placed within the target area and makes contact with the protagonist's bat simultaneously. Moreover, the antagonist will get 10 points of reward if the protagonist's bat is placed within the

target area, waiting for the antagonist to attack, for more than 2 seconds. Each game will last for 20 seconds. The observation space for both agents includes the Cartesian pose and velocity of the two bats, as well as the game time in seconds. For the sake of simplicity, both agents in this project are non-mobile. Yet, mobility can be easily integrated into future iterations of this game.

## 3 Related Work

**Reinforcement Learning in Competitive Games.** Competitive games have been used as benchmarks to evaluate the ability of algorithms to train an agent to make rational and strategic decisions [19, 20, 21]. Multi-agent reinforcement learning methods allow agents to learn emergent and complex behavior by interacting with each other and co-evolving together [22, 23, 24, 25, 26, 27]. Many recent efforts have used multi-agent RL methods to learn continuous control polices that can achieve high complexity tasks. Bansal et al. [28] created control policies for simulated humanoid and quadrupedal robots to play competitive games such as soccer and wrestling. Lowe et al. [29] has extended DDPG [30] to multi-agent settings by employing a centralized action-value function.

**Human-Robot Competition.** There are a few studies focusing on human-robot interaction in competitive games. For instance, Kshirsagar et al. [31] studied the affect of "co-worker" robots on human performance when they are performing in the same environment and competing for a monetary prize. This study showed that humans were slightly discouraged when competing against a high performing robot. On the other hand, humans exhibited a positive attitude towards the low performing robot. Mutlu et al. [32] showed that male subjects were more engaged in competitive video games when they played with an ASIMO robot. However, the majority of the subjects preferred cooperative games when they played with this robot. Short et al. [33] analysed the "rock-paper-scissors" game and found that human subjects were more socially and mentally engaged when the robot cheated during the game.

**Robots in Physical Training.** In the context of using robots to assist humans in physical exercises, a robot developed by Fasola and Mataric [34] was able to provide real-time coaching and encouragement for a seated arm exercise. Süssenbach et al. [35] developed a motivational robot for indoor cycling exercise that employed a set of communication techniques in response to the subject's physical condition. The results showed that the robot sufficiently increased the users' workout efficiency and intensity. Sato et al. [36] created a system capable of imitating the motion and strategy of top volleyball blockers for assisting vollyball training.

These works are typically limited to a few competitive scenarios that require simple and repetitive motions. In this work, we leverage reinforcement learning to create a robotic system that can potentially play various physically competitive games against human players.

## 4 System Design and Implementation

Humans are highly efficient at recognizing patterns and learning skills from just a small number of examples [37]. Existing research also shows that human subjects can adapt to robots and improve their performance in just a few trials in physical HRI tasks [38, 39]. However, games against a robot competitor are less challenging if a human player can easily predict its behavior and quickly find an optimal counter-strategy. We hypothesized that human players can quickly learn to improve their performance against a given robot policy, but a change in the robot's gameplay style could interrupt this learning effect and keep the games challenging. A gameplay style is characterized by patterns in the agent's end-effector motion trajectories. For example, one antagonist may prefer using more stabbing movements, while another antagonist may prefer slashing movements. Therefore, there are two primary objectives that govern the generation of robot control policies. First, a policy should allow the robot to play the Fencing Game sufficiently well, such that the games are intense and engaging to the human users. Second, we propose to obtain three policies with unique gameplay styles for our user study. A multi-agent reinforcement method created the robot control policies that are used in the user study. We designed and implemented the physical system based on a PR2 robot. Extra discussions on the physical system implementation and the technical details of the learning algorithm can be found in Appendix A.2.

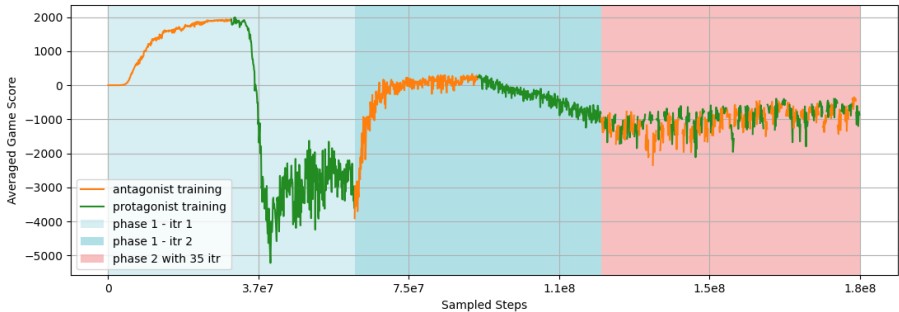

Figure 2: The antagonist's average game score during one complete round of phase one and two training. The two iterations of phase one training enabled two agents to interact according to the game rules. The phase two training performed 35 small updates resulting in random characteristics for both agents. Appendix A.2 contains extra interpretation for this figure.

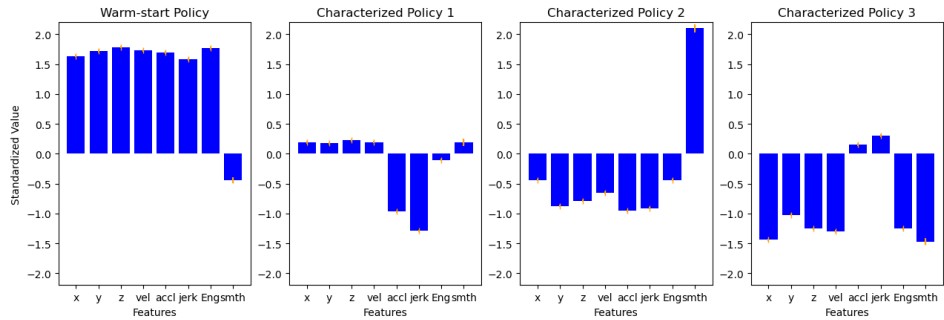

Figure 3: Visualization of quantified gameplay style of the four policies used in the user study. Error bars indicate the standard deviation of the feature values among the population.

## 4.1 Learning to Compete

To generate robot control policies that comply with the two aforementioned requirements, we formulate the Fencing Game as a multi-agent Markov game problem [40]. Both the antagonist agent and the protagonist agent are represented by a PR2 robot model in a Mujoco simulation environment. Both agents are trained in a co-evolving manner by playing games against each other. We break down the multi-agent proximal policy optimization method proposed by Bansal et al. [28] into two separate training processes, which reduces the computational processing needed to obtain multiple pairs of agents with acceptable performance. As a result, we were able to complete all sampling and training processes on a desktop computer (cpu: $1 \times i7$, gpu: $1 \times GTX970$). Our version of multi-agent PPO, the two-phase iterative co-evolution algorithm, is presented in Algo. 1.

**Learning to Move and Play.** The first phase of the training can be seen as a pre-training process, which aims to allow both agents to quickly learn the motor skills required for joint control and the rules of the game. The agents are rewarded by both the weighted sum of a continuous reward and the game score at each timestep. The continuous reward encourages the agents' exploration in the task space. The policy of the antagonist $\mu$ with parameters $\theta_i^\mu$ will first be trained by collecting trajectories that result from playing against the protagonist with its most recent policy. This process continues until timeout or $\mu$ has converged. The protagonist's policy $\nu$ with parameter $\theta_i^\nu$ will then be trained against the antagonist with its most recent policy. We acquired robot policies that exhibited competent (but still imperfect) gameplay by only running this training sequence twice ($N_{iter} = 2$). The pair of resulting policies from phase one will be called the warm-start policies.

**Creating Characterized Policies.** It has been shown that by using different random seeds to guide policy optimization toward different local optima, an agent can learn different behaviors and approaches to complete the same task [41, 26]. The highly variable nature of multi-agent systems enhances this random effect. The agents in multi-agent systems are more likely to learn drastically

different strategies and emergent behaviors when they continuously learn by competing with each other [28, 42, 43, 44]. The second phase of training generates a policy with random characteristics by exploiting this fact. Both agents are initialized with their warm-start policies from phase one and trained in the same iterative scheme shown in Algo. 1. Now the agents are solely rewarded by the game scores. When training each of the agents, instead of having an agent face against the opponent's latest policy, one of the previous versions of the opponent's policy in the history will be randomly selected in phase two. We created a policy library that contains six pairs of randomly characterized policies that resulted from six different rounds of phase two training. Fig. 2 demonstrates the change of game scores during the two-phase training process.

## 4.2 Selecting Agents With Distinct Gameplay Styles

In order to identify the most distinctive protagonist policies in the library described in the last subsection, each agent's gameplay style needs to be quantified and compared. We first generate trajectories for all protagonist policies in a tournament [45], where each agent plays 100 games with each of the six opponents. Eight end effector trajectory features are selected to quantify agent's gameplay styles: total displacement change on $x$, $y$, and $z$ axis, average velocity, average acceleration, average jerk, total kinetic energy, and trajectory smoothness. These features are calculated for each of the games played by each of the protagonists. The quantified style of a protagonist agent is the averaged features across 600 games. The features of all protagonists are then compared via their three most significant principal components (less than 2% of information loss)[46], and the three most separable policies are selected for the user study. The protagonist's gameplay style for the warm-start policy and the three selected characterized policies are visualized in Fig. 3.

---

**Algorithm 1:** Iterative Co-evolution

**Input:** Environment $\mathcal{E}$; Stochastic policies $\mu$ and $\nu$; Instantaneous Reward Function $r(\cdot)$
**Initialize:** Parameters $\theta_0^\mu$ for $\mu$ and $\theta_0^\nu$ for $\nu$
**for** $i = 1, 2, ..N_{iter}$ **do**
  **if** *phase one training* **then**
    $\theta_i^\nu \leftarrow \theta_{i-1}^\nu$
  **else**
    $\theta_i^\nu \leftarrow \theta_{random\_from\_history}^\nu$
  **end**
  **for** $j = 1, 2, ..N_\mu$ **do**
    rollout $\leftarrow roll(\mathcal{E}, \mu_{\theta_i^\mu}, \nu_{\theta_{i-1}^\nu}, r(\cdot))$
    $\theta_i^\mu \leftarrow$ PPO_Update(rollout)
  **end**
  **if** *phase one training* **then**
    $\theta_i^\mu \leftarrow \theta_i^\mu$
  **else**
    $\theta_i^\mu \leftarrow \theta_{random\_from\_history}^\mu$
  **end**
  **for** $j = 1, 2, ..N_\nu$ **do**
    rollout $\leftarrow roll(\mathcal{E}, \mu_{\theta_i^\mu}, \nu_{\theta_i^\nu}, r(\cdot))$
    $\theta_i^\nu \leftarrow$ PPO_Update(rollout)
  **end**
**end**

---

## 4.3 Sim2Real

Due to the kinematic and dynamic mismatch between simulation and reality, policies that are trained solely on simulated data can perform poorly in reality. We use a combination of a Jacobian Transpose end-effector controller and a system identification (systemID) process to solve this problem. Instead of specifying the torque values for each joint, the policy outputs an offset from the current end-effector pose. The new desired pose is executed by the end-effector controller. We used the CMA-ES algorithm to optimize the following objective over the parameter space of both the controller and the robot model in the simulation.

$$(\theta_m^*, \theta_c^*) = \arg \min_{(\theta_m, \theta_c)} \sum_{t=0}^{T} (s_r^t - s_s^t)^2$$

Where $\theta_m$ represents the simulated robot model parameters: damping, armature, and friction loss. $\theta_c$ represents the proportional gains and derivative gains of the end-effector controller. $T_r$ and $T_s$ are trajectories sampled from the real robotic system and simulation respectively that result from the same control sequence. $s_r^t \in T_r$ and $s_s^t \in T_s$ are the robot's end-effector pose in reality and simulation at time $t$ respectively. As a result, the difference in end-effector dynamics between the simulated robot and the real PR2 robot is reduced. The maximum controller output is bounded conservatively to prevent possible human injury.

Table 1: Subjective Ratings of the Modified TAM Questions

| | 1-Strongly Disagree | 2-Disagree | 3-Neutral | 4-Agree | 5-Strongly Agree |
|---|---|---|---|---|---|
| Perceived Usefulness | 0% | 6.25% | 25% | 37.5% | 31.25% |
| Perceived Ease of Use | 0% | 18.75% | 18.75% | 62.5% | 0% |
| Attitude | 0% | 6.25% | 37.5% | 25% | 31.25% |
| Intention to Use | 0% | 18.74% | 18.75% | 25% | 37.5% |
| Perceived Enjoyment | 0% | 6.25% | 6.25% | 31.25% | 56.25% |
| Desirability | 0% | 6.25% | 12.5% | 31.25% | 50% |
| Increase Engagement | 6.25% | 6.25% | 31.25% | 18.75% | 37.5% |

# 5   Experiments and Analysis

We performed two in-lab user studies in this work. The first user study performed a broad exploration on the idea of competitive-HRI under the fencing game setting. Sixteen human subjects were asked to play five games with each of the four RL trained policies that resulted from Sec. 4. Subjects' game scores, heart rates, arm movements, and their responses to a modified technology acceptance model (TAM)[47] were used to evaluate our system from the following three perspectives: 1. Is a competitive robot accepted by human users under the scenarios of physical games and exercise? 2. Can our system effectively create challenging and intense gameplay experience? 3. Can the robot interrupt the human learning effect by switching its gameplay style?

The second user study compared characterized policy 1 with a carefully designed heuristic-based policy. By placing its bat in between the target area and the point on the human's bat that is closest to the target area, the robot exploits embedded knowledge of the game's rules in order to execute a strong baseline heuristic policy. Action noise was added to the heuristic policy to create randomness in the robot's behavior. Ten human subjects were asked to play 10 games with each robot policy. This experiment compared the two policies via game scores, subjects' TAM responses, subjects' perception of difficulty, enjoyment, and robot intelligence. Details about the heuristic baseline policy design, experiment procedures, subjective question design, and the demographic information for both user studies are discussed in Appendix A.3.

## 5.1   User Study One Result: A Broad Exploration

Our first experiment demonstrates that participants highly accept the use of a competitive robot as an exercise partner. The majority of the subjects considered competitive games with our robot to be useful, entertaining, desirable, and motivating. Our system was able to provide a challenging and intensive interactive experience that significantly increased subjects' heart rates. While competing against our RL trained robot, most subjects struggled to significantly improve their performance over time. Yet, a subset of subjects who constantly explored different strategies achieved higher scores in the long run.

**User Acceptance.** Table 1 summarizes the subjects' responses to the technology acceptance model. The majority of the subjects (*i.e.* 68.75%) agreed that a competitive robot could improve the quality of their physical exercise. Moreover, 87.5% of subjects agreed that competitive human-robot interactive games are entertaining, and 81.25% of subjects agreed that a competitive robot exercise companion would be desirable in the future. Interestingly, the intention to use (62.5% agree + strongly agree) and increased engagement (56.25% agree + strongly agree) metrics are not as high as the vast agreement on perceived enjoyment and desirability. Some subjects who didn't have a strong intention to use our system explained their reasons in the open-ended question. They stated that it is not immediately clear how competitive robots can play a role in their routine exercises, such as jogging, and weight training. On the other hand, most subjects (*i.e.*, 71%) who exercise less than three hours per week agreed that a competitive robot partner will increase their engagement with physical exercise. Therefore, our competitive robot is more effective in motivating people who do relatively little exercise. Future research should explore how to effectively apply competitive-HRI to common exercises.

**Increased Heart Rate.** Most of the gameplay with our competitive robot increased the human subjects' heart rates significantly. Subjects' peak heart rates were higher than their resting heart rates in more than 99% of the games, and their peak heart rates in 92.6% of the games were higher than their walking baseline heart rates. Since the subjects were asked to keep their feet planted on the ground during a game, their body maneuverability was very limited. Despite this movement limitation, subjects' peak heart rates were significantly higher (*i.e.*, 20% to 58%) than their walking baseline in 29% of the games. We found that, other than physical effort, subjects' cognitive effort and reported emotions also corresponded to a rise in heart rate. Figure 4. b. shows that sections with higher average heart rate corresponded to people feeling cognitive demand, motivated, frustrated, and intimidated by the robot. In short, playing competitive games with our robot can be cognitively demanding and can trigger noticeable emotional reactions.

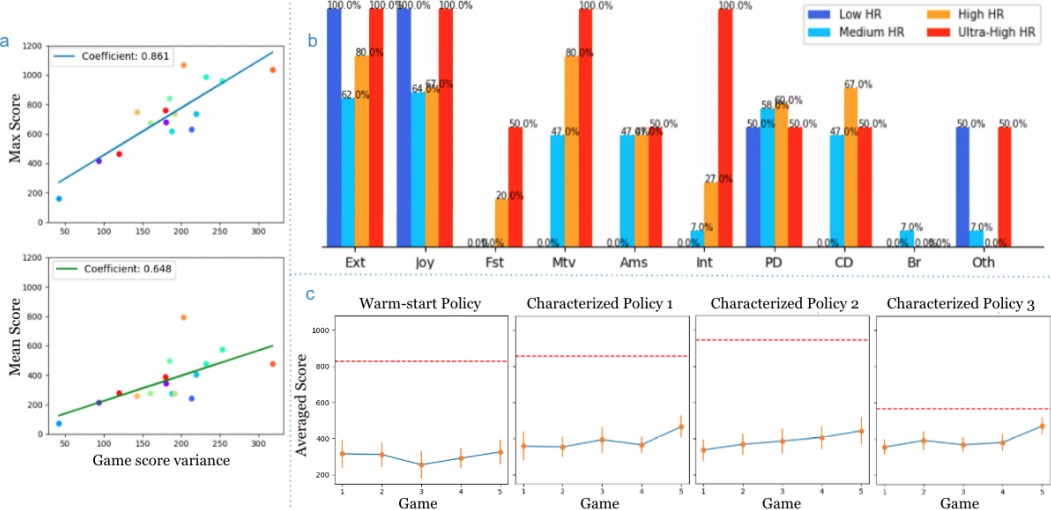

Figure 4: **a.** The 16 points in each subplot represent the 16 subjects. The variance of subjects' game scores is positively correlated to their achieved maximum and mean scores. **b.** Comparison of subjective descriptions between sections with low, medium, high, and ultra-high averaged heart rates. The definition of each group is detailed in Appendix A.5. The x-axis of each subplot shows the adjective describing each section of games (Exciting, Joyful, Frustrating, Motivating, Amusing, Intimidating, Physically Demanding, Cognitively Demanding, Boring, Others). The y-axis indicates the percentage of subjects in the corresponding group who selected the corresponding answer. **c.** Each subplot shows the average game scores of all subjects on the five games against each robot policy. The error bars indicate the standard errors over the samples. The red horizontal lines indicate the best mean score achieved by one of the subjects against the corresponding robot policy.

**The Human Learning Effect.** As mentioned in Sec. 4, we wanted to test if changing the robot's gameplay style interrupts the human learning process and keeps the interaction challenging throughout the whole experiment. This hypothesis was based on our assumption that most human players can make significant improvement within five games against a fixed robot policy. Surprisingly, this assumption was invalidated by our user study data. Fig. 4. c. shows the average game scores and standard deviation of all subjects in the five consecutive games against each of the four robot policies. An analysis of variance (ANOVA) suggests that subjects have no significant performance increase between the five games for a given policy (Warm-start Policy: $p = 0.96$, Characterized Policy 1: $p = 0.74$, Characterized Policy 2: $p = 0.85$, Characterized Policy 3: $p = 0.43$). The red horizontal dashed line in each subplot of Fig. 4. c. represents the best mean score achieved by a subject, which approximates the performance of an observed best human policy. The average performance of all subjects were lower than (*i.e.*, much lower in most cases) the performance of the best human policies, and most subjects still have much room for performance improvement. Our assumption on human learning was based on evidence from noncompetitive HRI experiments. In contrast, our competitive setting creates a much more dynamical environment and resulted in a more challenging learning environment for the human. Since no significant performance variance was observed across the four sections either (ANOVA $F = 1.51, p = 0.26$), our system was able to

remain challenging to users throughout the whole experiment. However, more studies are needed to analyze how changes in gameplay style interrupt human learning.

**Human Performance.** Despite the fact that no significant learning effect was found in the subject population, we observed that some subjects tried multiple strategies in the experiment, which could be interpreted as exploration within a learning framework. This observation motivated us to analyze the relationship between strategy exploration and performance, which we first quantified as variance in game scores, and then as the featurized gameplay styles from Sec. 4. As shown in Fig 4. a., a larger variance in score corresponds to better performance in terms of maximum and mean scores. We then compared the variance of the gameplay style between the five subjects with the highest maximum scores and the five subjects with the lowest maximum scores. For the sake of simplicity, we only compared the end-effector displacement change on $x$, $y$, and $z$ axes and the averaged velocity. The variance of each selected feature for subjects with high performance were at least 29% higher than subjects with low performance. This suggested that subjects who have high variance in score tend to take risks by constantly exploring different strategies, resulting in better rewards in the long run. Future research could examine whether a robot can help human users achieve better performance by verbally or implicitly encouraging them to explore different strategies.

## 5.2 User Study Two Result: Baseline Comparison

This experiment compared an RL trained policy (characterized policy 1) to a strong heuristic baseline policy in a long gameplay sequence setting (10 games per policy). Compared to the baseline, the RL trained policy has a slightly better game score performance. In contrast to the first experiment, the human learning effect was observed in both policies. However, the RL trained policy was significantly better at suppressing human subjects from learning to make progress even without switching gameplay styles during the experiment. While the responses to the TAM model were very similar for both policies, the subjects considered the RL policy to be more intelligent because of its "defensive" and "diverse" behavior.

**Game Scores:** When playing against the baseline policy, the subject population's average game scores (**Baseline mean:** 383.5, **RL mean:** 349.1), the maximum game score (**Baseline max:** 929.0, **RL max:** 744.0), and the minimum game score (**Baseline min:** -291.0, **RL min:** -320.0) are higher than those against the RL policy. Instead of switching policies every five games, we used a longer sequence of game-plays to evaluate each policy in this study. We were able to find a positive correlation between game scores and the amount of game-play experience against a specific policy. A linear regression over the **Baseline** method data (slope=30.0, coefficient=0.56, p value=9.3e-10) has a larger positive slope, stronger correlation coefficient, and a smaller statistical significant p value compared to that of the **RL policy** (slope=15.7, coefficient=0.34, p value=0.0005).

**Subjective Responses:** Subjects' responses to most of the modified TAM questions for the two policies are very similar as shown in Table 3. However, 70% of the population considered the **RL policy** to be more intelligent. In the responses for short questions 2, 3, and 4 in Table 2, the **Baseline** policy was described as "fast" by 2 subjects, "follows my movement" by 4 subjects, and has "repetitive/predictable" behavior by 4 subjects. Meanwhile, the **RL** policy was considered to be "defensive" by 5 subjects, "strategic" by 2 subjects, and to have "diverse behavior" by 2 subjects.

## 6 Conclusion

This work motivated research in competitive-HRI by discussing how competition can be beneficial to people and the technical difficulties competitive-HRI tasks represent. The Fencing Game, a physically interactive zero-sum game is proposed to evaluate system capability in certain competitive-HRI scenarios. We created a competitive robot using an iterative multi-agent RL algorithm, which is able to challenge human users in various competitive scenarios, including the Fencing Game. Our first user study found that human subjects are very accepting of a competitive robot exercise companion. Our competitive robot provides entertaining, challenging, and intense gameplay experiences that significantly increases the subject's heart rate. In our second user study, one of the policies resulting from the proposed RL method was compared to a strong heuristic baseline policy. The RL-trained policy were significantly better at suppression of the human learning affect, and appeared to be more intelligent to 70% of the population.

**Acknowledgments**

This work was supported in part by NSF awards EFMA-1832795 and CNS-1305072. This study (IRB ID: STUDY00012211) has been approved by the University of Washington Human Subjects Division.

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
