# OpenReview forum: "Motivating Physical Activity via Competitive Human-Robot Interaction"
_robot-learning.org/CoRL/2021/Conference — CoRL2021 Oral_

### Official Review · Reviewer_BAQR · 2021-07-20

**Originality:** Excellent
**Technical Quality:** Excellent
**Clarity Of Presentation:** Very Good
**Impact:** 4

**Recommendation:**

Strong Accept: I recommend accepting the paper and will argue for my recommendation even if other reviewers hold a different opinion.

**Summary:**

The authors develop a competitive fencing game to motivate humans to exercise while playing against a robot partner. To make the game interesting, the authors had to use state of the art robotics techniques such as sim2real and population-based training to produce a set of proficient and diverse policies. The experimental results showed that the human subjects enjoyed the game and did get some exercise, quantified by an increase in heart rate.

**Issues:**

I noticed at least a few typos:
Line 33 heading should be “System Design and Implementation”
Line 202 heading could be “Experiments”


**Reviewer Expertise:**

Good: General knowledge of the area

**Strengths And Weaknesses:**

The authors provide a brief but very interesting literature review about how competition improves motivation for humans performing a variety of physical and cognitive tasks. I’m surprised that there hasn’t been more work in the field of competitive HRI. It seems like a natural fit to make physical therapy more fun for patients.

The explanation of the scoring in section 2 was hard to understand, but watching the video linked in the caption of Figure 1 was very helpful. From what I understand, the video has a high pitched beep when the human scores points, and a low pitched beep when the robot scores (when the foam rods touch).

The authors develop a fencing game that is enjoyable and physically demanding.
The robot opponent’s capabilities must be sufficient for the game to be challenging and interesting, so the authors explain the technical aspects of the development of the robot's fencing policy. The authors first use self-play in a Mujoco simulation to train a protagonist and antagonist. The authors use population based training to encourage diversity of policies.
The authors select the 3 most distinctive policies out of the population using PCA analysis of trajectory roll-outs. The authors effectively bridge the Sim2Real transfer using the CMA-ES algorithm. A key part of the approach is that the robot switches between 3 different policies, so that the game is more challenging because the robot is less predictable.

The explanation of the experimental methods was very clear. It is good that the authors were able to obtain baseline heart rate data as a control for the study. Also, the test subjects had 5 minutes to learn about the scoring of the game before the start of scoring.
The results of the survey are impressive, especially in the perceived enjoyment category of the survey.
The finding about the changes in emotional and cognitive reactions was also very interesting.
It was surprising that humans were not able to improve at the game over 5 episodes without the use of the diverse policies. Maybe the game is more challenging than expected. The human players did explore several strategies.


**Summary Of Recommendation:**

The paper is very well motivated, extremely novel, and has great technical execution and experimental methods & results.

---

> ### Author Response · Authors · 2021-08-31
> **Response to Reviewer BAQR**
>
> We want to first thank the reviewers for their time and helpful inputs to our paper. In addition to the original experiment, we performed a second user study with 10 human subjects that compared one of our RL trained policies (the characterized policy one) to a carefully designed heuristic baseline policy in a long gameplay sequence setting (10 consecutive games per policy). Compared to the baseline, the RL trained policy has a better game score performance. In contrast to the first experiment, the human learning effect was observed in both policies. However, the RL trained policy was significantly better at suppressing human subjects from learning to make progress even without switching gameplay styles during the experiment. While the responses to the TAM model were similar for both policies, the subjects considered the RL policy to be more intelligent because of its "defensive" and "diverse" behavior.
>
> Re >> “The explanation of the scoring in section 2 was hard to understand” /  “I noticed at least a few typos: Line 33 heading should be “System Design and Implementation” Line 202 heading could be ‘Experiments’”
>
> Thank you for pointing out these problems, we’ve created an algorithm that describes the scoring mechanism in Appendix A.1 to help readers better understand the game rules. Typos are fixed in the revised manuscript.

---

### Official Review · Reviewer_DiX9 · 2021-07-23

**Originality:** Good
**Technical Quality:** Poor
**Clarity Of Presentation:** Good
**Impact:** 3

**Recommendation:**

Weak Reject: I recommend rejecting the paper, but will not argue for my recommendation if the majority of other reviewers have a different opinion.

**Summary:**

This paper presents a novel human-robot interaction task — the Fencing Game — with a focus on building a system for this task that encourages healthy *human-robot competition*, an area that has historically been understudied in the HRI literature.

Generally, the goal of the system is to build a robot capable of “fencing” (simplified, safe, no real swords) with a human for encouraging physical exercise. In addition to defining the rules of this two-player zero-sum game in which a human needs to attempt to move their sword to a “target area” without contacting the robot’s sword (straightforward attack/defense style gameplay), this paper introduces a system for learning diverse robot policies for playing this game, trained in simulation via a variant of self-play, then transferred to the real world.

The system is evaluated with a user study that measures quantitative metrics like increase in users heart rate while playing the game (proxying the efficacy in a physical exercise setting) and average game score, as well as quantitative metrics like enjoyment, usefulness, ease of use, etc. (adapted from the standard set of technology acceptance model questions).

A crucial part of the proposed approach is building a “catalogue” of diverse robot policies for the humans to play against; the idea here is that diversity prevents humans from “overfitting” too much to a given policy and maximizes the sustained efficacy of such a system. However, I found this part of the paper hard to understand (when are different policies being shown to users during the study), and the results do not seem to back up the hypothesis, which I found interesting.

**Issues:**

There are several missing details in the learning section of the paper that also contribute to my recommendation above. Specifically:

- What is the continuous reward used to optimize the self-play policies (beyond game score)? Where does this come from?
- Could you provide more details about the “diversity” phase of training - are you just continuing training from the same “warm-start” point with different randomness?
- How do you shortlist the 12 policies you add to the “diversity catalogue”
- How would the straightforward baseline (phase 1 training, but for longer) perform?
- How does the diverse class of 3 policies show up in the user study? Do you switch policies on a user within the 5 games they play, or do you do this across users.
- Generally, would appreciate a lot more clarity around the user study setup!

**Reviewer Expertise:**

Very good: Comprehensive knowledge of the area

**Strengths And Weaknesses:**

The biggest strength of this paper is the exploration of competitive human-robot interaction, and formalizing such a paradigm through this fencing game. I believe this is an interesting axis for human-robot interaction, and I hope to see more work like this in the future!

However, I have serious concerns about the experimental procedure, and weaknesses in the study, and the corresponding conclusions. Notably:
- There is no basis for comparing the proposed learned robot policy. I have no idea how good these human scores are, and looking at qualitative metrics like “ease of use” and “perceived usefulness” without any point of comparison makes it hard to draw any conclusions.
- For the given setup of the game, it seems like a random heuristic that just waves the bat back and forth really quickly across the target area would provide a substantial challenge for humans, while giving them a similar workout (but perhaps at the cost of “enjoyment”). A random baseline where you inject some noise into this heuristic might actually considerably beat the learned baselines.
- The process for learning robot policies via self-play is interesting, but it’s again hard to see 1) why the multi-phase learning process is necessary (especially since the “warm-start” policy performs as well as the other diverse policies), and 2) how diversity in policies is actually being quantified/the guiding principle that defines this part of the learning process.

---
**Update Post-Rebuttal**

The authors did a lovely job addressing my concerns during the rebuttal. I still have some questions regarding the multi-phase learning policy and would love to see more of an ablation/intuitive explanation of the policy diversity, but the basis of comparison question has been answered with the heuristic. I am updating my score to a weak reject -- while I think the comparison is good, I think there's still room to expand the set of experiments to fully study these questions (I'm still curious about a fully random baseline, or a baseline that just moves the bat back and forth quickly).

**Summary Of Recommendation:**

In its current form, this paper tackles a well-motivated question in building a system for healthy human-robot competition. However, while the initial part of the paper is well-motivated, there are several concerns around the learning and evaluation involved in the proposed system that give me significant reservations.

Specifically:
- Lack of critical details and motivation around some of the learning algorithm design choices, and in how the user study was run.
- [Critical] Lack of a basis for comparison - the proposed system seems technically sound, but I have no idea how to evaluate its efficacy — without this, it’s hard to even argue that this particular game (the Fencing Game) is the *right* paradigm for studying human-robot competition. Notably, there are simple heuristic/random baselines that I think could easily be implemented that would outperform the existing system.

---
**Update Post-Rebuttal**

See above - I am pleased by the rebuttal and updating my score to a weak reject.

---

> ### Author Response · Authors · 2021-08-31
> **Response to Reviewer DiX9**
>
> We want to first thank the reviewers for their time and helpful inputs to our paper. In response to the comment from reviewers 1 and 2 that there was no comparison to a baseline hand-coded (non-learned) policy, we have performed a second user study with 10 human subjects. The second study compares one of our RL-trained policies ( “characterized policy” one) to a carefully designed heuristic baseline policy in a long gameplay sequence setting (10 consecutive games per policy). Compared to the baseline, the RL trained policy has a better game score performance. In contrast to the first experiment, the human learning effect was observed in both policies. However, the RL-trained policy made it significantly more challenging for subjects to make quantitative improvements even without switching gameplay styles during the experiment. While the responses to the TAM model were similar for both policies, the subjects considered the RL policy to be more intelligent because of its "defensive" and "diverse" behavior.
>
> Re >> “There is no basis for comparing the proposed learned robot policy” / “Lack of a basis for comparison” / “Notably, there are simple heuristic/random baselines that I think could easily be implemented that would outperform the existing system”
>
> Originally, we were having trouble finding the most sensible baseline to compare to. A benefit of a self-supervised learning approach is that it has the potential to enable generalization, unlike a hard-coded policy, which can only play the specific game it was coded for. By modifying the simulation environment and the reward functions, the proposed method allows the robot to learn to play potentially any physically competitive games. Therefore, the ideal baseline should also be a generalized robotic system/method that can play competitive games, like the fencing game, without requiring significant modification or extensive manual tuning (such as tuning a heuristic function). Yet, to the best of our knowledge, such a directly comparable system does not exist. Since the proposed system in this work is the first of its kind, we believe that the experimental data generated by this system provides good technical insights and can be a good baseline for future competitive-HRI research.
>
> However, we do agree that having a baseline robot that is not created under the iterative RL training scheme allows us to better understand the proposed method. We started by testing a random policy but found that this will almost certainly make the game uninteresting to the user. Therefore we’ve decided not to use such a weak baseline because we want to have a more interesting comparison. We instead implemented a strong baseline by encoding expert knowledge into a heuristic function with a degree of motion randomness. We performed an extra user study and found that the two policies are different in both quantitative metrics (game scores, and suppression on the human learning effect) and a qualitative metric (subjective descriptions of the robot behavior). The baseline policy design details and the new experimental results are discussed in Appendix A3 and Section 5 respectively in the revised manuscript.

---

> > ### Author Response · Authors · 2021-08-31
> > **Response to Reviewer DiX9 Part 2**
> >
> > Re >> “why the multi-phase learning process is necessary (especially since the “warm-start” policy performs as well as the other diverse policies)” / “How would the straightforward baseline (phase 1 training, but for longer) perform?”
> >
> > This is a great question that we should have clarified in the paper! There are two main differences between phase one and phase two training:
> > 1. Phase one training uses a continuous reward to facilitate agents’ development of basic motor skills. In phase two training, the agents are solely rewarded by the game scores.
> > 2. In phase one training, each agent is always learning against the latest version of the opponent. But in phase two training, an agent would constantly and randomly load a previous version of the opponent from the history, after a short period of training.
> >
> > The use of continuous reward encourages the robot to quickly explore the task space, which makes the phase one training good for quickly initializing the policies for the antagonist and protagonist. However, continued use of the iteration strategy in phase one training would likely trap both agents in a low-quality local equilibrium and/or chasing each other in circles in parameter space in a long sequence of training[1]. In contrast, the reward and iteration mechanisms of phase two training create a high variance learning environment which effectively mitigates the circling problem [2, 3]. In addition, because of the high variance nature of the phase two training, agents are more likely to learn emergent behaviors and converge to more sophisticated policies as described in Section 4.1. Indeed, in Appendix A5 (or A6 in the original version of the paper), we showed that subjects’ perceived enjoyment and difficulty were positively correlated to the amount of data used to train the corresponding policy. Subjects considered the characterized policies (trained with approximately 1.8e8 steps of data) to be more enjoyable and difficult to play with compared to the warm-start policy (trained with approximately 1.2e8 steps). We added clarification on this topic in Appendix A2 on the revised paper.
> >
> > [1] Mertikopoulos, Panayotis, Christos Papadimitriou, and Georgios Piliouras. "Cycles in adversarial regularized learning." Proceedings of the Twenty-Ninth Annual ACM-SIAM Symposium on Discrete Algorithms. Society for Industrial and Applied Mathematics, 2018.
> > [2] Bansal, Trapit, et al. "Emergent complexity via multi-agent competition." arXiv preprint arXiv:1710.03748 (2017).
> > [3] Lowe, Ryan, et al. "Multi-agent actor-critic for mixed cooperative-competitive environments." arXiv preprint arXiv:1706.02275 (2017).
> >
> > Re >> “how diversity in policies is actually being quantified/the guiding principle that defines this part of the learning process.” / “Could you provide more details about the “diversity” phase of training - are you just continuing training from the same “warm-start” point with different randomness?”
> >
> > As discussed in the previous section, diversity in the learned characterized policies is completely driven by the high degree of randomness used in phase two training. The behavioral style of each resulting agent was quantified by their trajectories features as described in Section 4.2.
> >
> > Re >> “What is the continuous reward used to optimize the self-play policies (beyond game score)? Where does this come from?”
> >
> > For the antagonist, the continuous reward encourages it to shorten the distance between its sword and the target area. For the protagonist, the continuous reward encourages it to follow the antagonist’s sword movement.
> >
> > Re >> “How do you shortlist the 12 policies you add to the ‘diversity catalogue’ ”
> >
> > As discussed in sections 4.1 and 4.2, we have six pairs of randomly characterized policies that resulted from six different rounds of phase two training. Note that the fencing game is an asymmetric game, such that the protagonist agents and antagonist agents are not interchangeable. We only focused on the protagonist agents because the antagonist agents are used to simulate the human players in a simulation environment (and therefore will not be executed on the real robot). We then sample trajectories for all protagonist policies in a tournament and quantify their behavior styles with eight end-effector trajectory features. The three most separable policies in the feature space are selected for the user study.
> >
> > Re >> “How does the diverse class of 3 policies show up in the user study?”
> >
> > Each user plays five consecutive games against each of the three characterized policies. Before the experiment with each subject, the robot will randomly order the three characterized policies and load a policy once every five games according to this order. For example, a randomly generated order -- (3, 1, 2) will have the subject to first play five consecutive games with the characterized policy 3, then five consecutive games with the characterized policy 1, and five games with the characterized policy 2.

---

> ### Comment · Reviewer_DiX9 · 2021-09-03
> **Update Post-Rebuttal**
>
> Thanks to the authors for a lovely rebuttal. I have updated my score to a Weak Reject, since I still have some reservations about the comparison, and some of the explanations around the multi-phase training procedure, and policy diversity.

---

### Official Review · Reviewer_h8aJ · 2021-07-23

**Originality:** Very Good
**Technical Quality:** Very Good
**Clarity Of Presentation:** Excellent
**Impact:** 4

**Recommendation:**

Strong Accept: I recommend accepting the paper and will argue for my recommendation even if other reviewers hold a different opinion.

**Summary:**

This work shows that a PR2 trained to fence against another PR2 can used the learned policy to compete with humans, challenging them with a Fencing Game to encourage physical activity. The paper shows that not only were the robots able to compete physically with human users, the users enjoyed the experience. This work is a good start for a “robot workout buddy” that becomes competitive through training with other robots, and then encourages people to workout using competitive games.

**Issues:**

Issues to address during the author response/revision period:
-	Is there any data recorded during training, etc. to suggest that the robot’s training improved its gameplay and potentially human enjoyment over an untrained robot?
-	Was there a significant visually observable difference between the distinct gameplay styles?
-	What metrics were used to determine that the Fencing Game was played “sufficiently well” after two training sequences?


**Reviewer Expertise:**

Very good: Comprehensive knowledge of the area

**Strengths And Weaknesses:**

This paper is well written and clear. The idea of competitive robotic partners to encourage exercise is really interesting, and it seems like the execution worked quite well. I appreciate that this work trained agents with different gameplay styles, and also tested whether people would learn how to play against the robot quickly and become bored. The results of this paper seem to support the claim that a robot can train to compete with a person in a physical game, and that people enjoy playing against the robot.

While the results show that the system worked and was enjoyable for people, it could have been useful to see baselines to compare against. Maybe measuring enjoyment/performance for people playing against a robot moving randomly, rather than with training, to show that the robot was able to learn a better policy than random movements? Or compare enjoyment/human gameplay performance against playing the Fencing Game with another person instead of a robot?

Overall, this paper is interesting and makes a strong argument for competitive robots that learn how to play games in a challenging way

**Summary Of Recommendation:**

I am recommending a weak accept because this paper is interesting and makes a strong argument for competitive robots that learn how to play games in a challenging way. However, it does not have baselines to compare against to show that the training was necessary to challenge the users. Overall I think this is interesting, well written work.


Edit post-rebuttal: The paper was updated during the rebuttal period to include a baseline comparison for the proposed system, which changed my assessment of the technical quality. This experimental addition was helpful to understanding the system, and has changed my recommendation to a strong accept.

---

> ### Author Response · Authors · 2021-08-31
> **Response to Reviewer h8aJ**
>
> We want to first thank the reviewers for their time and helpful inputs to our paper. In response to the comment from reviewers 1 and 2 that there was no comparison to a baseline hand-coded (non-learned) policy, we have performed a second user study with 10 human subjects. The second study compares one of our RL-trained policies ( “characterized policy” one) to a carefully designed heuristic baseline policy in a long gameplay sequence setting (10 consecutive games per policy). Compared to the baseline, the RL trained policy has a better game score performance. In contrast to the first experiment, the human learning effect was observed in both policies. However, the RL trained policy made it significantly more challenging for subjects to make quantitative improvements even without switching gameplay styles during the experiment. While the responses to the TAM model were similar for both policies, the subjects considered the RL policy to be more intelligent because of its "defensive" and "diverse" behavior.
>
> Re >> “it could have been useful to see baselines to compare against. Maybe measuring enjoyment/performance for people playing against a robot moving randomly, rather than with training, to show that the robot was able to learn a better policy than random movements? ”
>
> A benefit of a self-supervised learning approach is that it has the potential to enable generalization, unlike a hard-coded policy, which can only play the specific game it was coded for. By modifying the simulation environment and the reward functions, the proposed method allows the robot to learn to play potentially any physically competitive game. Therefore, the ideal baseline should also be a generalized robotic system/method that can play competitive games, like the fencing game, without requiring significant modification or extensive manual tuning (such as tuning a heuristic function). However, to the best of our knowledge, such a directly comparable system does not exist. Since the proposed system in this work is the first of its kind, we believe that the experimental data generated by this system provides good technical insights and can be a good baseline for future competitive-HRI research.
>
> However, we do agree that having a baseline robot that is not created under the iterative RL training scheme allows us to better understand the proposed method. We started by testing a random policy but found that this will almost certainly make the game uninteresting to the user. Therefore we’ve decided not to use such a weak baseline because we want to have a more interesting comparison. We instead implemented a strong baseline by encoding expert knowledge into a heuristic function with a degree of motion randomness. We performed an extra user study and found that the two policies are different in both quantitative metrics (game scores, and suppression on the human learning effect) and a qualitative metric (subjective descriptions of the robot behavior). The baseline policy design details and the new experimental results are discussed in Appendix A3 and Section 5 respectively in the revised manuscript.
>
> Re >> “Is there any data recorded during training, etc. to suggest that the robot’s training improved its gameplay and potentially human enjoyment over an untrained robot?”
>
> Yes, there is. In Appendix A5 (or A6 in the original version of the paper), we showed that subjects’ perceived enjoyment and difficulty were positively correlated to the amount of data used to train the corresponding policy. Subjects considered the characterized policies (trained with approximately 1.8e8 steps of data) to be more enjoyable and difficult to play against compared to the warm-start policy (trained with approximately 1.2e8 steps).
>
> Re >> “Was there a significant visually observable difference between the distinct gameplay styles?”
>
> During the user study, various subjects expressed seeing/feeling the difference between different agents, yet these comments were not recorded officially.
>
> Re >> “What metrics were used to determine that the Fencing Game was played “sufficiently well” after two training sequences?”
>
> The average game scores of the sampled trajectories, as shown in Figure 2 in the paper, were used to determine the status of the two training agents. At the beginning of the training (i.e., phase 1 - itr 1) the game scores tend to be largely biased toward whichever agent is currently learning. This is because both agents’ policies are simple/naive at the beginning phase and the opponent can easily find a counter strategy to dominate the game during learning. After the warm-start training (i.e., phase 1 - itr 2), we see the game scores converge to an equilibrium where both agents are challenging each other without completely dominating the game. These training interpretation details are also added to the revised manuscript. We also observed the agents’ gameplay in simulation as a qualitative evaluation.

---

> ### Comment · Reviewer_h8aJ · 2021-09-02
> **Updated Response**
>
> The authors answered my questions, and addressed the concerns about the lack of a baseline by adding new results to the paper. I have updated my review to a strong accept.

---

### Meta-Review · Area_Chair_z8tX · 2021-08-14

**Recommendation:** Accept (Oral)
**Confidence:** 5

**Metareview:**

### Final Meta-Review

Reviewers unanimously express their appreciation of the authors' effort and thoroughness in responding to their concerns during the discussion and revision phase. The authors' inclusion of a new user study to measure differences between the proposed learning-based approach and a hand-crafted policy has been well received. Nonetheless, reviewer DiX9 maintains some unresolved concerns, which the authors should endeavor to address in the final version of the paper, by discussing possible confounding factors and limitations of the user study.

All in all, the final manuscript has been significantly strengthened and, given the overall novelty and appeal of its ideas, I am happy to recommend it for inclusion in the CoRL proceedings and Oral Presentation at the conference.

### Original Meta-Review

The paper proposes a Fencing Game as a human-robot competition setup meant to encourage physical exercise. The reviewers agree that the paper is well written and explores an interesting direction, making a compelling argument for this type of application.

Two of the three reviewers express concerns with the experimental evaluation of the proposed system: in particular, without a control group of participants engaging with a baseline robot policy, it is unclear how much of the objective and subjective results should be attributed specifically to the proposed learned approach. This is issue is accentuated by the little justification provided for some of the (presumably heuristic) design choices, as well as some missing technical details.

The authors should thoroughly address the reviewers' concerns and questions during the response period, especially those regarding experimental evaluation and design decisions.

---

> ### Author Response · Authors · 2021-08-31
> **Response to Area Chair z8tX**
>
> In response to the reviewers' biggest concern that there was no comparison to a baseline hand-coded (non-learned) policy, we have performed a second user study with 10 human subjects. The second study compares one of our RL-trained policies ( “characterized policy” one) to a carefully designed heuristic baseline policy in a long gameplay sequence setting (10 consecutive games per policy). Compared to the baseline, the RL trained policy has a better game score performance. In contrast to the first experiment, the human learning effect was observed in both policies. However, the RL trained policy made it significantly more challenging for subjects to make quantitative improvements even without switching gameplay styles during the experiment. While the responses to the TAM model were similar for both policies, the subjects considered the RL policy to be more intelligent because of its "defensive" and "diverse" behavior. We also provided comprehensive answers to reviewers' questions and concerns over multiple topics such as "baseline selection", "efficacy of the proposed training method", "simple heuristic", and "technical details". Please see our direct response to each of the reviewers for more details.
>
> We revised our manuscript based on the reviewers' inputs, the major modifications including:
> - Section 5 includes the experimental results from the new user study.
> - Further technical details and system design chooses are discussed in Appendix A.2
> - Training process interpretation is also discussed in Appendix A.2
> - Extra user study information, such as the detailed experimental procedure for the two user studies, questionnaire, design details of the baseline heuristic policy are explained in Appendix A.3.

---

### Decision · Program_Chairs · 2021-09-13

**Decision:**

Accept (Oral)

**Comment:**

### Final Meta-Review

Reviewers unanimously express their appreciation of the authors' effort and thoroughness in responding to their concerns during the discussion and revision phase. The authors' inclusion of a new user study to measure differences between the proposed learning-based approach and a hand-crafted policy has been well received. Nonetheless, reviewer DiX9 maintains some unresolved concerns, which the authors should endeavor to address in the final version of the paper, by discussing possible confounding factors and limitations of the user study.

All in all, the final manuscript has been significantly strengthened and, given the overall novelty and appeal of its ideas, I am happy to recommend it for inclusion in the CoRL proceedings and Oral Presentation at the conference.

### Original Meta-Review

The paper proposes a Fencing Game as a human-robot competition setup meant to encourage physical exercise. The reviewers agree that the paper is well written and explores an interesting direction, making a compelling argument for this type of application.

Two of the three reviewers express concerns with the experimental evaluation of the proposed system: in particular, without a control group of participants engaging with a baseline robot policy, it is unclear how much of the objective and subjective results should be attributed specifically to the proposed learned approach. This is issue is accentuated by the little justification provided for some of the (presumably heuristic) design choices, as well as some missing technical details.

The authors should thoroughly address the reviewers' concerns and questions during the response period, especially those regarding experimental evaluation and design decisions.